

# Camostat mesilate inhibits pro-inflammatory cytokine secretion and improves cell viability by regulating MFGE8 and HMGN1 in lipopolysaccharide-stimulated DF-1 chicken embryo fibroblasts

Lin Yuan[1],Mengjie Li[2],Zhishuai Zhang[3],Wanli Li[1],Wei Jin[1] and Mingfa Wang[1]

[1] Henan Key Laboratory of Farm Animal Breeding and Nutritional Regulation, Institute of Animal Husbandry and Veterinary Medicine, Henan Academy of Agricultural Sciences, Zhengzhou, Henan, China
[2] Bureau of Agriculture and Rural Affairs of Longting District, Kaifeng, Henan, China
[3] Henan Institute of Animal Health Supervision, Zhengzhou, Henan, China

Corresponding author
Mingfa Wang,
wangmingfa@hnagri.org.cn

## ABSTRACT

Camostat mesilate (CM) possesses potential anti-viral and anti-inflammatory activities. However, it remains unknown whether CM is involved in lipopolysaccharide (LPS)-mediated inflammatory responses and cell injury. In this project, differentially expressed proteins (DEPs, fold change $\geq$ 1.2 or $\leq$ 0.83 and Q value $\leq$ 0.05) in response to LPS stimulation alone or in combination with CM were identified through tandem mass tags (TMT)/mass spectrometry (MS)-based proteomics analysis in DF-1 chicken embryo fibroblasts. The mRNA expression levels of filtered genes were determined by RT-qPCR assay. The results showed that CM alleviated the detrimental effect of LPS on cell viability and inhibited LPS-induced TNF-α and IL-6 secretions in DF-1 chicken embryo fibroblasts. A total of 141 DEPs that might be involved in mediating functions of both LPS and CM were identified by proteomics analysis in DF-1 chicken embryo fibroblasts. LPS inhibited milk fat globule EGF and factor V/VIII domain containing (MFGE8) expression and induced high mobility group nucleosome binding domain 1 (HMGN1) expression, while these effects were abrogated by CM in DF-1 chicken embryo fibroblasts. MFGE8 knockdown facilitated TNF-α and IL-6 secretions , reduced cell viability, stimulated cell apoptosis in DF-1 chicken embryo fibroblasts co-treated with LPS and CM. HMGN1 loss did not influence TNF-α and IL-6 secretions, cell viability, and cell apoptosis in DF-1 chicken embryo fibroblasts co-treated with LPS and CM. In conclusion, CM exerted anti-inflammatory and pro-survival activities by regulating MFGE8 in LPS-stimulated DF-1 chicken embryo fibroblasts, deepening our understanding of the roles and molecular basis of CM in protecting against Gram-negative bacteria.

## INTRODUCTION

Camostat mesilate (CM, Foipan<sup>TM</sup>), a protease inhibitor, has been used for the treatment of multiple diseases such as chronic pancreatitis and recessive dystrophic epidermolysis bullosa in human (*McKee et al., 2020*; *Ramsey, Nuttall & Hart, 2019*; *Ikeda et al., 1988*). Moreover, some studies have proposed that CM (an inhibitor of TMPRSS2 serine protease) might act as a candidate drug against SARS-CoV-2 and COVID-19 given its inhibitory effect on host cell entry of SARS-CoV-2 and its protective effects against SARS-CoV-induced mouse death (*McKee et al., 2020*; *Uno, 2020*; *Hoffmann et al., 2020*). In addition, CM has potential anti-viral activities against influenza and parainfluenza viruses (*Yamaya et al., 2015*; *Hosoya et al., 1992*). Furthermore, previous *in vitro* and *in vivo* studies have demonstrated that CM exerts anti-inflammatory activity in LPS-activated rat monocytes and chronic pancreatitis and genetically diabetic rat models (*Gibo et al., 2005*; *Jia, Taguchi & Otsuki, 2005*). For instance, Jia et al. have pointed out that CM can inhibit the expression of interleukin (IL)-1β, tumor necrosis factor-α (TNF-α), and IL-6 in the fibrotic or degenerative regions of the pancreas in obese and diabetic rats (*Jia, Taguchi & Otsuki, 2005*). However, few studies have been performed currently to investigate the association of CM and Gram-negative bacteria-mediated inflammatory responses and cell injury.

Gram-negative bacteria, a group of pathogens infecting all eukaryotes, can trigger multitudinous diseases (*e.g.*, infertility and fowl cholera) in birds including chickens (*Deb, Chatturvedi & Jaiswal, 2004*; *Harper & Boyce, 2017*; *Neyen & Lemaitre, 2016*). Moreover, birds carrying bacteria can transmit pathogens to other animals and human, which seriously influences public health and food safety (*Gargiulo et al., 2018*; *Abebe, Gugsa & Ahmed, 2020*; *Nga et al., 2019*). Toxins and virulent factors generated by bacteria play vital roles in the determination of the pathogenesis of these pathogens (*Abebe, Gugsa & Ahmed, 2020*). Lipopolysaccharide (LPS), also known as endotoxin, is one of the main virulent factors of most Gram-negative bacteria (*Neyen & Lemaitre, 2016*). It has reported that the toll-like receptor (TLR) 4 signaling pathway serves as a crucial player in the response to extracellular LPS or bacteria (*Mazgaeen & Gurung, 2020*; *Brownlie & Allan, 2011*). For instance, the expression levels of TLR4, TLR4 pathway-related genes (*e.g.*, myeloid differentiation primary response 88, TNF receptor associated factor 6, interferon regulatory factor 7) and inflammatory cytokines (*e.g.*, IL-2, IL-6, IL-1β, and TNF-α) have been found to be dysregulated in peripheral mononuclear blood cells (PBMCs) of Aseel, Ghagus, Dahlem red, and Broiler chickens upon LPS treatment (*Karnati et al., 2015*). Also, the allelic variation in TLR4 has been reported to be associated with the resistance to Salmonella enterica serovar Typhimurium infection in chickens (*Leveque et al., 2003*). Fibroblasts, a kind of antigen-presenting cell, can respond to the signals related to injuries and pathogens (*e.g.*, bacteria, viruses) by TLRs including TLR4 (*Turner et al., 2018*; *Pinheiro et al., 2018*). For example, *Zhao et al. (2013)* have demonstrated that the expression levels of TLR4 and TLR4 downstream genes (IL-6, MHC II, and IL-1β) are dysregulated in duck embryo fibroblasts following LPS stimulation. Moreover, not only live bacteria but also LPS can induce pro-inflammatory responses and subsequent cell injury in fibroblasts (*Hao et al., 2017*; *Lian et al., 2018*). Fibroblasts, one member of the

connective-tissue cell family, function as vital players in the development of multiple diseases such as organ damage/fibrosis (*Turner et al., 2018*; *Darby & Hewitson, 2007*) and arthritis (*Croft et al., 2019*). Moreover, fibroblasts have been identified as the sentinel cells (cells that can transform into pro-inflammatory phenotype when required) in some infectious diseases such as keratitis (*Fukuda et al., 2017*) and periodontitis (*Baek, Choi & Ji, 2013*; *Scheres et al., 2011*). Additionally, fibroblasts are involved in the initiation, modulation, and maintenance of inflammation (*Enzerink & Vaheri, 2011*).

In this text, the effects and molecular mechanisms of CM on Gram-negative bacteria-induced inflammation and cell injury were preliminarily explored by ELISA assay, CCK-8 assay and tandem mass tags (TMT)/mass spectrometry-based proteomics analysis in LPS-treated DF-1 chicken embryo fibroblasts. Moreover, we further investigated whether high-mobility group (HMG) nucleosome binding domain 1 (HMGN1) and milk fat globule epidermal growth factor 8 (MFGE8) were involved in the regulation of CM functions in LPS-treated DF-1 chicken embryo fibroblasts.

## MATERIALS AND METHODS

### Cell culture
DF-1 chicken embryo fibroblasts (CRL-12203; American Type Culture Collection, Manassas, VA, USA) were cultured in Dulbecco's Modified Eagle Medium (Thermo Scientific, Waltham, MA, USA) containing 10% fetal bovine serum (Thermo Scientific) at 39 °C, and treated with different concentrations of LPS (Sigma-Aldrich, St. Louis, MO, USA) for 6 h.

### Reagents
Lipopolysaccharides (from Escherichia coli) (L2630) were purchased from Sigma-Aldrich. CM was obtained from ChemeGen Lnc. (Los Angeles, California, USA). Small interference RNAs (siRNAs) targeting MFGE8 (si-MFGE8#1 and si-MFGE8#2), HMGN1 (si-HMGN1#1 and si-HMGN1#2), and a scrambled siRNA control (si-NC) were synthesized by GenePharma Co., Ltd. (Shanghai, China). Cells were transfected with corresponding siRNAs using Lipofectamine 2000 reagent (Thermo Scientific) according to the instructions of the manufacturer. SiRNA sense sequences were presented in Table S1.

### RT-qPCR assay
Total RNA was extracted from DF-1 chicken embryo fibroblasts using Trizol reagent (Thermo Scientific). Next, RNA was reversely transcribed into cDNA first strand using M-MLV Reverse Transcriptase (Thermo Scientific) following the manufacturer's instructions. The mRNA expression levels of MFGE8, myristoylated alanine-rich protein kinase C substrate (MARCKS), HMGN1, high mobility group nucleosomal binding domain 4 (HMGN2), and GAPDH were measured through the real-time quantitative system using cDNA template, SYBR Green PCR Master Mix (Thermo Scientific) and corresponding quantitative primers. The

primer sequences were presented as below: 5′-CGATCTGAACTACATGGTTTAC-3′(forward) and 5′-TTCACTCTGATGCGGTCCAC-3′(reverse) for GAPDH, 5′-GCCACGTCAAAGACTGGAAAC-3′(forward) and 5′-AATGCCTGCACCCCCTTATAC-3′(reverse) for HMGN1, 5′-TGTTAGCACACAGACCGCTT-3′(forward) and 5′-TTCACTCTGATGCGGTCCAC-3′(reverse) for HMGN2, 5′-TTACCACCATTCCAACGGGC-3′(forward) and 5′-GATCCCTTATCGACCCACCC-3′(reverse) for MARCKS, 5′-CAAGGTTTTCCAGGGCAACG-3′(forward) and 5′-GCAACCTGCCGTGTTGAAAT-3′(reverse) for MFGE8.

## ELISA assay

DF-1 chicken embryo fibroblasts transfected with or without si-NC, si-MFGE8, or si-HMGN1 were treated with LPS for 6 h and then incubated with CM for an additional 24 h. TNF-α and IL-6 secretion levels were measured using corresponding ELISA kits (MEIMIAN Biotechnology Co., Ltd., Wuhan, China) following the manufacturer's protocols.

## CCK-8 assay

Cell viability was determined through Cell Counting Kit-8 (Beyotime Biotechnology, Shanghai, China) according to the manufacturer's protocols. Briefly, cells were inoculated into 96-well plates. At the indicated time points after treatment, 10 μl of CCK-8 solution was added into each well (100 μl medium/well). Three hours later, the optical density (OD) values were determined at 450 nm.

## Cell apoptosis detection

Cell apoptotic rate was determined by Annexin V-FITC Apoptosis Staining/Detection Kit (Solarbio, Beijing China). Briefly, DF-1 cells were collected and re-suspended in 500 μl of 1 x Binding Buffer. Then, cells were co-incubated with 5 μl of Annexin V-FITC and 5 μl of PI at room temperature in a dark room. Cell apoptotic patterns were analyzed by flow cytometry (Becton Dickinson Co. CA, USA).

## TMT-based proteomics analysis

Cell samples (three biological replicates) were lysed using the protein lysis buffer, sonicated, and centrifuged to obtain cell lysates containing proteins. After treated with dithiothreitol (DTT) and iodoacetamide (IAM), proteins were digested into peptide segments with trypsin. Next, the peptide segments post enzymolysis were labeled with TMT and fractionated on an L-3000 high-performance liquid chromatography (HPLC) system. Finally, fractions were analyzed and detected using EASY-nLCTM 1200 Ultra High-Performance Liquid Chromatography (UHPLC, Thermo Scientific) and Q Exactive HF-X Mass Spectrometer (Thermo Scientific). Proteins were identified using Proteome Discoverer 2.2 (Thermo Scientific). Proteins were quantified using Maxquant software. GO and KEGG pathway enrichment analyses were performed using the KOBAS 3.0 website (http://kobas.cbi.pku.edu.cn/kobas3).

## Statistical analysis

In the proteomics analysis, differences between groups were analyzed using Student's $t$-test. $P$ values obtained from Student's $t$-test were corrected through false discovery rate (FDR).

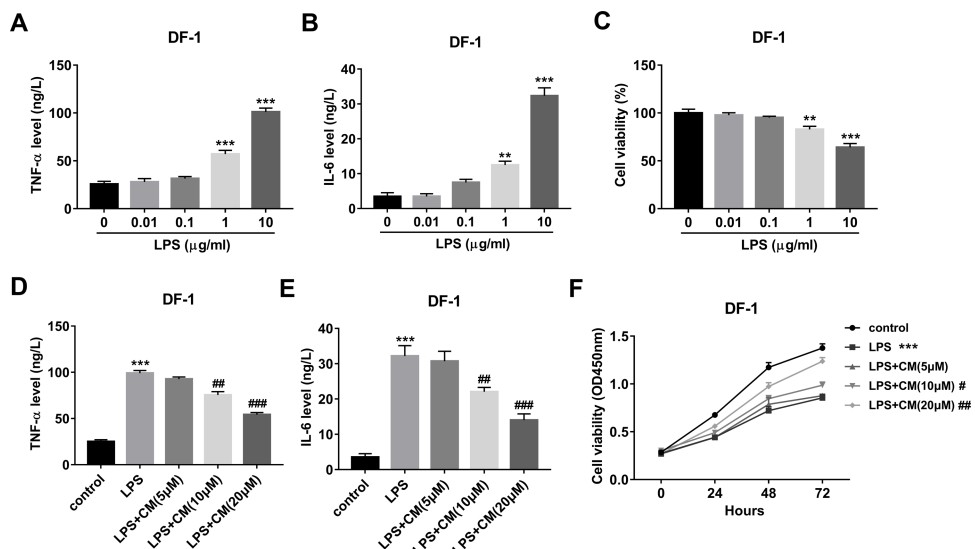

**Figure 1** (A-F) CM protected DF-1 chicken embryo fibroblasts from LPS-induced inflammation and cell injury.

The *P* values after FDR correction were termed as Q values. Proteins were regarded as differentially expressed at the fold change ≥1.2 or ≤0.83 and Q value ≤0.05.

In the cell experiments, data analysis was conducted using GraphPad Prism software 7.0 (La Jolla, CA, USA) with the results presenting as mean ± standard deviation. Differences between groups were compared using Student's *t*-test. Differences among groups were analyzed through one-way ANOVA (Tukey's post-hoc test) or two-way ANOVA (Sidak post hoc test). Differences were regarded as statistically significant when the *P* value was less than 0.05.

## RESULTS

### CM weakened LPS-induced pro-inflammatory cytokine secretion and cell injury in DF-1 chicken embryo fibroblasts

Firstly, our data revealed that LPS facilitated TNF-α and IL-6 secretions and decreased cell viability in a concentration-dependent manner with the strongest effect at the concentration of 10 μg/ml (Figs. 1A–1C). Hence, 10 μg/ml of LPS was used to construct the cell model. Moreover, our study demonstrated that CM dose-dependently inhibited the secretions of TNF-α and IL-6 in the LPS (10 μg/ml)-exposed DF-1 cell model (Figs. 1D and 1E). Additionally, CM increased cell viability in a time- and concentration-dependent manner in LPS-treated DF-1 chicken embryo fibroblasts (Fig. 1F). These data suggested that CM could protect DF-1 chicken embryo fibroblasts from LPS-induced inflammation and cell injury.

## Identification of dysregulated proteins in response to LPS stimulation alone or in combination with CM in DF-1 chicken embryo fibroblasts

Next, dysregulated proteins in DF-1 chicken embryo fibroblasts upon the treatment of LPS alone or in combination with CM were screened out by TMT/mass spectrometry (MS)-based proteomics analysis. Results showed that 273 proteins were differentially expressed (182 up-regulated, 91 down-regulated) in LPS treated DF-1 cell group (DL group) compared to the untreated cell group (D group) (Figs. 2A and 2B). Moreover, 102 proteins were highly expressed and 547 proteins were low expressed in DF-1 chicken embryo fibroblasts co-treated with LPS and CM (DLE group) than that in DF-1 chicken embryo fibroblasts treated with LPS alone (DL group) (Figs. 2A and 2B). In addition, 26 common proteins were identified by Venn analysis among down-regulated proteins in DL *vs* D group and up-regulated proteins in DLE *vs* DL group (Fig. 2A). Also, 115 proteins were found to be up-regulated in DL *vs* D group and down-regulated in DLE *vs* DL group (Fig. 2B). These proteins with the converse change trends in DLE *vs* DL and DL *vs* D groups might be involved in the responses of DF-1 chicken embryo fibroblasts to LPS and CM. The heat map of the proteins with converse change trends in DLE *vs* DL and DL *vs* D groups was presented in Fig. S1 and corresponding protein information was displayed in Table S2. Next, GO and KEGG enrichment analysis for these proteins with the converse change trends in DLE *vs* DL and DL *vs* D groups were carried out to decrypt the protective mechanisms of CM in LPS-stimulated DF-1 chicken embryo fibroblasts. The top 20 GO enrichment terms based on biological processes, cellular components, and molecular functions were presented in Figs. 2C–2E. GO enrichment analysis showed that these proteins were mainly involved in the regulation of biological processes such as cellular/metabolic/biological processes, cellular component organization or biogenesis, gene expression, protein/nitrogen compound/peptide/amide transport, and apoptotic cell clearness (Fig. 2C, Table S3). KEGG enrichment analysis revealed that 7 KEGG pathways (ribosome, steroid biosynthesis, RNA transport, MAPK signaling pathway, spliceosome, herpes simplex virus 1 infection, and salmonella infection) were significantly enriched by these proteins (Fig. 2F, Table S4). These enrichment analyses suggested that CM could trigger the notable dysregulation of a host of essential genes related to cell development.

## Effects of LPS alone or in combination with CM on the expression of MFGE8, MARCKS, HMGN1, and HMGN2 in DF-1 chicken embryo fibroblasts

Among proteins with the inverse alteration trends in DLE *vs* DL and DL *vs* D groups, 4 proteins (MFGE8 (*Yi, 2016*), MARCKS (*Green et al., 2011*), HMGN1 (*Yang et al., 2012*), and HMGN2 (*Xie et al., 2011*)) related to inflammation were screened out for further explorations according to previous documents. Annotation analysis revealed that MFGE8 was involved in the regulation of multiple pathways or biological processes such as protein metabolism, post-translation protein modification, insulin-like growth factor (IGF) transport and uptake by insulin-like growth factor binding proteins (IGFBPs), phagocytosis, and apoptotic cell clearance (Table S5). MARCKS participated in actin/protein kinase C/calmodulin binding, actin filament organization, central nervous system development,

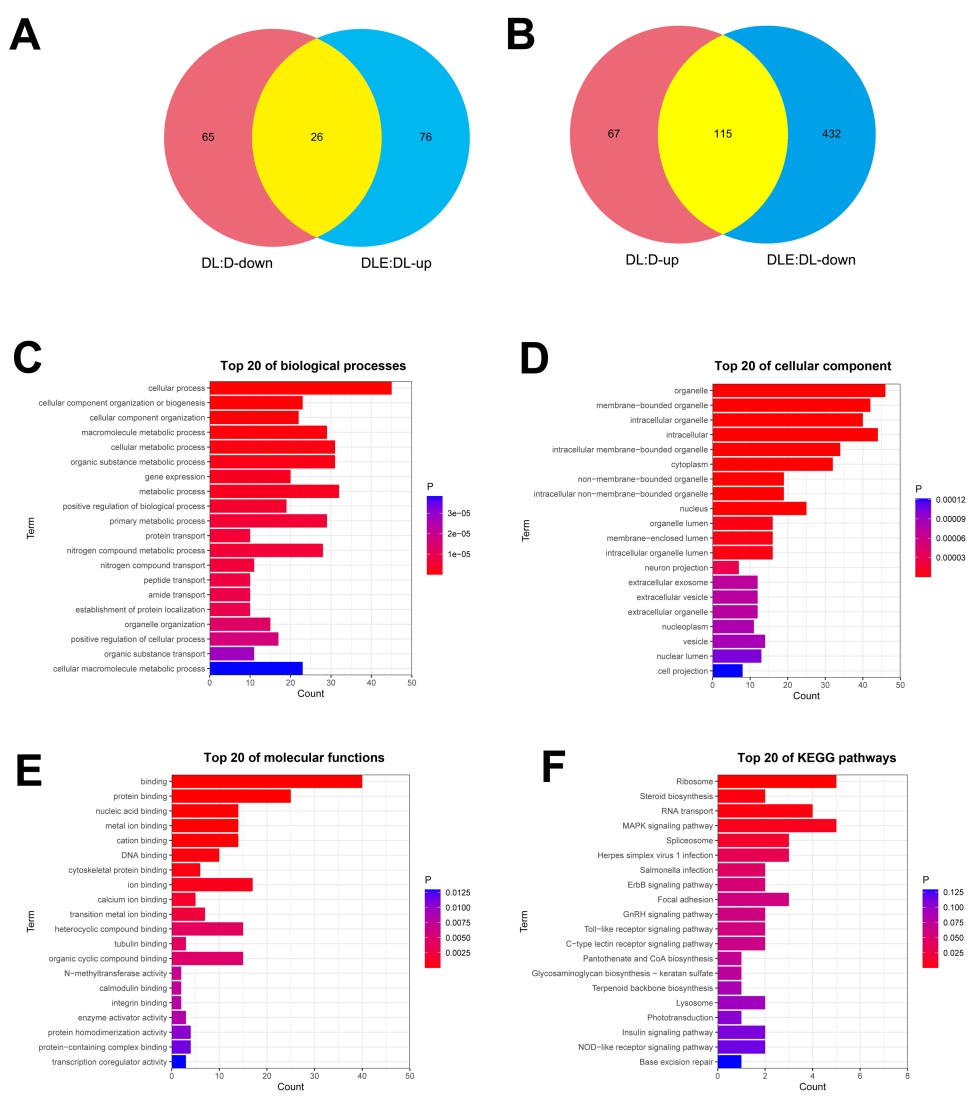

**Figure 2** (A–F) Identification of dysregulated proteins in response to LPS stimulation alone or in combination with CM in DF-1 chicken embryo fibroblasts.

and actin crosslink formation (Table S5). HMGN1 could regulate nucleotide-excision repair, chromatin organization, transcription by RNA polymerase II, UV-B/UV-C response, and NAD+ ADP-ribosyltransferase activity (Table S5). HMGN2 was implicated in chromatin organization (Table S5). TMT-MS data revealed that MFGE8 expression was notably increased in DLE *vs* DL group (1.66-fold) and markedly reduced in DL *vs* D group (0.81-fold) (Table S2). The fold-change values of MARCKS, HMGN1, and HMGN2 in the DLE *vs* DL group are 0.61, 0.59, and 0.55, respectively. The fold-change values of MARCKS, HMGN1, and HMGN2 in the DL *vs* D group are 1.39, 1.54, 1.89, respectively (Table S2). Consistent with the TMT-MS outcomes, RT-qPCR assay also disclosed that MFGE8 expression was markedly reduced, and HMGN1 and HMGN2 expression was notably increased in LPS-stimulated DF-1 chicken embryo fibroblasts (DL group) *versus*

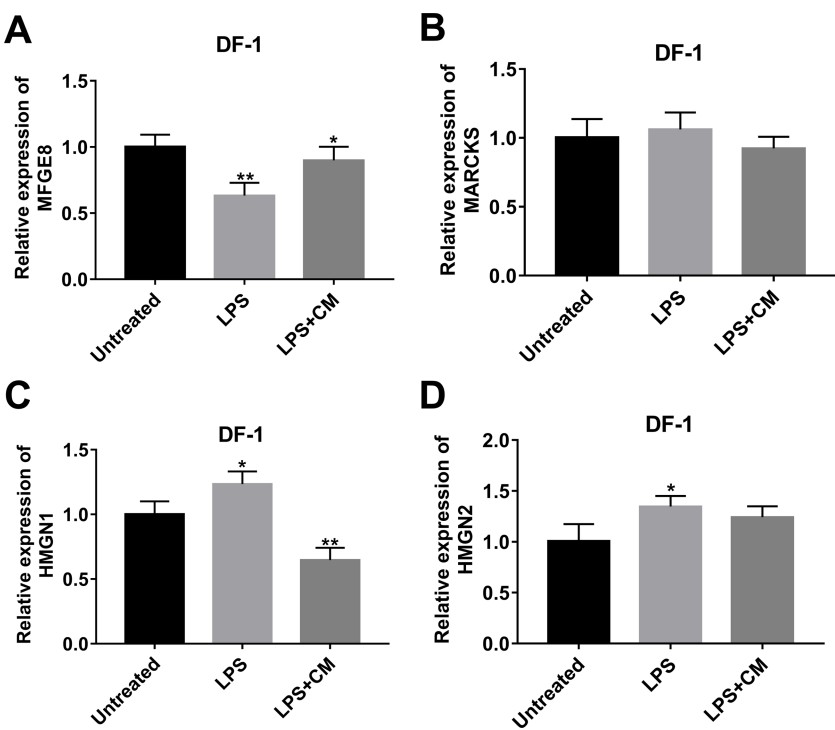

**Figure 3** Effects of LPS alone or in combination with CM on the expression of (A) MFGE8, (B) MAR-CKS, (C) HMGN1, and (D) HMGN2 in DF-1 chicken embryo fibroblasts.

un-stimulated cells (D group) (Figs. 3A, 3C, 3D). However, there was no obvious alteration in MARCKS expression in LPS-stimulated DF-1 chicken embryo fibroblasts compared to untreated cells (Fig. 4B). In line with TMT-MS data, higher MFGE8 expression and lower HMGN1 expression were also validated by RT-qPCR assay in DF-1 chicken embryo fibroblasts with the combined treatment of LPS and CM (DLE group) compared to cells treated with LPS alone (Figs. 3A, 3C). Nevertheless, the addition of CM did not influence MARCKS and HMGN2 expression in LPS-treated DF-1 chicken embryo fibroblasts (Figs. 3B, 3D). Given the consistency of RT-qPCR and proteomics data, HMGN1 and MFGE8 were selected for further investigations.

## CM exerted its functions through regulating MFGE8 and HMGN1 expression in LPS-treated DF-1 chicken embryo fibroblasts

To further investigate the functions of MFGE8 and HMGN1, si-NC and siRNAs targeting MFGE8 and HMGN1 were synthesized. Knockdown efficiency analyses revealed that the introduction of si-MFGE8#1 led to the notable down-regulation of MFGE8 mRNA level in DF-1 chicken embryo fibroblasts compared with the si-NC group, while si-MFGE8# 2 could not effectively silence MFGE8 in DF-1 chicken embryo fibroblasts (Fig. 4A). Also, a noticeable reduction of HMGN1 mRNA expression level was observed in DF-1 chicken embryo fibroblasts transfected with si-HMGN1#1 or si-HMGN1#2 than that in cells transfected with si-NC (Fig. 4B). Considering the stronger knockdown outcomes of

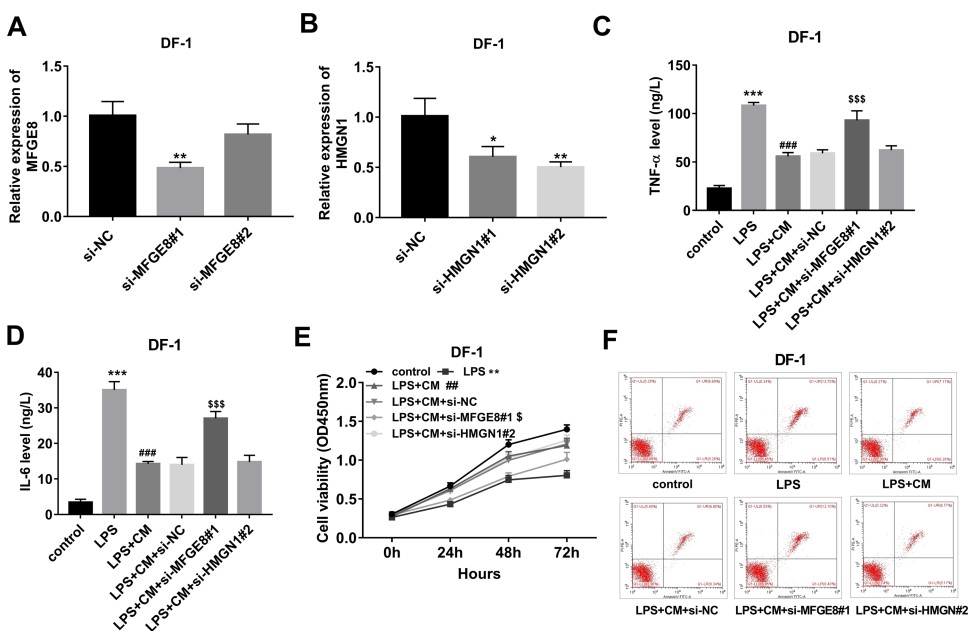

**Figure 4** (A-F) CM exerted its anti-inflammatory and pro-survival activities by regulating MFGE8 and HMGN1 expression in LPS-treated DF-1 chicken embryo fibroblasts.

si-MFGE8#1 and si-HMGN1#2 on their respective targets, si-MFGE8#1 and si-HMGN1#2 were selected in the subsequent loss-of-function explorations. Functional analyses revealed that MFGE8 knockdown could weaken the effects of CM on TNF-α and IL-6 secretions, cell viability, and cell apoptosis in LPS-treated DF-1 chicken embryo fibroblasts (Figs. 4C–4F). Moreover, our TMT-MS outcomes showed that CM treatment led to the notable increase of MFGE8 in LPS-treated DF-1 chicken embryo fibroblasts. These data suggested that CM could inhibit TNF-α and IL-6 secretions and improved cell viability by up-regulating MFGE8 in LPS-treated DF-1 chicken embryo fibroblasts. Additionally, HMGN1 knockdown did not have statistically significant influence on TNF-α and IL-6 secretions, cell viability and apoptosis in DF-1 chicken embryo fibroblasts co-treated with LPS and CM (Figs. 4C–4F).

## DISCUSSION

In this text, our data revealed that LPS stimulated the secretions of pro-inflammatory factors (TNF-α and IL-6) and reduced cell viability in DF-1 chicken embryo fibroblasts, suggesting the successful establishment of LPS-induced inflammation and cell injury model. Similarly, fibroblasts have been found to sense LPS and LPS stimulation led to the notable increase in the expressions of pro-inflammatory cytokines (*e.g.*, IL-1β, TNF-α, IL-8, and IL-6) in human gingival fibroblasts (*Kim et al., 2007*). LPS induced cell apoptosis and inhibited cell proliferation and migration in Guinea pig fibroblasts (*Gonciarz et al., 2019*).

Considering the anti-inflammatory activity of CM, the effects of CM on pro-inflammatory factor secretion and cell viability along with its molecular mechanisms were further investigated in LPS-stimulated DF-1 chicken embryo fibroblasts. Our results showed that CM inhibited the secretions of TNF-α and IL-6 and increased cell viability in LPS-stimulated DF-1 chicken embryo fibroblasts, suggesting that CM could protect chicken embryo fibroblasts from LPS-induced inflammation and cell damage.

Next, proteins that might be involved in mediating the anti-inflammatory and pro-survival activities of CM were screened out by TMT/MS-based proteomics analysis in the LPS-induced DF-1 cell models. Among proteins with the converse change trends in DLE *vs* DL and DL *vs* D groups, HMGN1 and MFGE8 were selected out to further investigate in virtue of the consistency of proteomics data and RT-qPCR outcomes.

HMGN1, formerly named as HMGN14, has been found to be implicated in multiple pathophysiologic processes such as inflammation, immunity, and fibrosis (*Yang et al., 2012*; *Wei et al., 2014*; *Yu et al., 2019*). For instance, *Yang et al. (2012)* demonstrated that HMGN1 stimulated pro-inflammatory cytokine (*e.g.*, IL-6, IL-8, TNF) production, facilitated dendritic cell maturation, induced leukocyte recruitment, and enhanced immune responses induced by ovalbumin or/and LPS in mice. Recombinant HMGN1 induced the production of pro-inflammatory cytokines such as IL-6, TNF-α, and IL-1 in human peripheral blood mononuclear cells (*Arts et al., 2018*). Given the close link between HMGN1 and inflammation, we further investigated whether CM could exert its functions by regulating HMGN1 expression in LPS-treated DF-1 chicken embryo fibroblasts. Our data revealed that HMGN1 knockdown or not did not influence TNF-α and IL-6 secretions, cell viability and apoptosis in DF-1 chicken embryo fibroblasts co-treated with LPS and CM.

Milk fat globule epidermal growth factor 8 (MFGE8), a peripheral membrane glycoprotein, also has been found to be involved in inflammation and immunity (*Yi, 2016*; *Li et al., 2013*). For instance, Nakaya et al. demonstrated that MFGE8 knockout enhanced apoptotic cell accumulation and inflammatory responses, and cardiac dysfunction in myocardial infarction mice (*Nakaya et al., 2017*). MFGE8 weakened LPS-induced pro-inflammatory responses in mouse macrophages, microglia, and brains (*Li et al., 2019*; *Aziz et al., 2011*). Our proteomics and RT-qPCR data showed that MFGE8 expression was notably reduced in LPS-treated DF-1 chicken embryo fibroblasts compared to untreated cells, but was markedly increased in DF-1 chicken embryo fibroblasts co-treated with LPS and CM compared to cells treated with LPS alone. Functional analyses revealed that MFGE8 knockdown weakened CM-mediated pro-survival and anti-inflammatory activities in LPS-treated DF-1 chicken embryo fibroblasts. Because CM is an inhibitor of TMPRSS2, the potential link between TMPRSS2 and MFGE8 was established using the STITCH website (http://stitch.embl.de/cgi/). Results showed that the possible association between TMPRSS2 and MFGE8 might be mediated by calcium ions (Fig. S2).

However, protein antibodies specific for chicken are relatively poor. The potential targets of CM identified by proteomics analysis and other experimental approaches need to be further validated. To examine the effect of CM on protein expression of CM targets in chicken cells, we need to align the protein sequences of CM targets between chicken and

human to examine whether protein antibody specific for human can be used in chicken. If the percentage of identical sequences reaches more than 80%, we would try to measure the protein levels of these targets in chicken using the antibody specific for human. In our project, the identity between human and chicken MFGE8 protein sequences was only about 50%. So, we did not detect the protein level of chicken MFGE8 using the antibody specific for human MFGE8 protein. Because the identity of human and chicken MFGE8 protein sequences is relatively low, the alterations that occurred in human cell lines cannot represent the effects in chickens.

Taken together, our data revealed that CM alleviated the detrimental effect of LPS on cell viability and weakened LPS-induced pro-inflammatory responses partly by regulating MFGE8 and HMGN1 expression in chicken embryo fibroblasts. These data suggested that CM might help chicken embryo fibroblasts fight against Gram-negative bacteria infection by regulating MFGE8, deepening our understanding of the roles and protective mechanisms of CM in Gram-negative bacteria-induced inflammatory responses and cell injury and hinting at the potential value of CM in protecting cells from Gram-negative bacteria. Moreover, a multitude of proteins that might be involved in regulating DF-1 cell responses to Gram-negative bacterium infection alone or in combination with CM treatment were screened out by TMT-based proteomics analysis.

### Funding

This work was supported by Science-Technology Foundation for Outstanding Young Scientists of Henan Academy of Agricultural Sciences (Grant no. 2020YQ20) and the Project of Science and Technology of the Henan Province for Tackling Key Problems (Grant no. 212102110181). The funders had no role in study design, data collection and analysis, decision to publish, or preparation of the manuscript.

### Grant Disclosures

The following grant information was disclosed by the authors:
Science-Technology Foundation for Outstanding Young Scientists of Henan Academy of Agricultural Sciences: 2020YQ20.
Project of Science and Technology of the Henan Province for Tackling Key Problems: 212102110181.

### Competing Interests

The authors declare there are no competing interests.

### Author Contributions

- Lin Yuan conceived and designed the experiments, performed the experiments, authored or reviewed drafts of the paper, and approved the final draft.
- Mengjie Li and Zhishuai Zhang analyzed the data, prepared figures and/or tables, and approved the final draft.

- Wanli Li and Wei Jin performed the experiments, authored or reviewed drafts of the paper, and approved the final draft.
- Mingfa Wang conceived and designed the experiments, authored or reviewed drafts of the paper, and approved the final draft.

## Data Availability

The detailed experimental procedures of TMT-based proteomics analysis are available in the Supplemental Files.

## Supplemental Information

Supplemental information for this article can be found online at http://dx.doi.org/10.7717/peerj.12053#supplemental-information.

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
