# Peer review of "Camostat mesilate inhibits pro-inflammatory cytokine secretion and improves cell viability by regulating MFGE8 and HMGN1 in lipopolysaccharide-stimulated DF-1 chicken embryo fibroblasts"

_PeerJ, doi:10.7717/peerj.12053_

## Round 0.1 · original submission · Major Revisions

The authors demonstrate the anti-inflammatory and pro-survival effect of Camostat mesilate (CM) in LPS-stimulated DF-1 cells accompanied by the altered protein expressions of MFGE8 and HMGN1.

The result is definitely meaningful and interesting. However, the current paper is required to be actively revised according to the reviewers' comments before it could be considered for publication in PeerJ. Please carefully read the reviewers' comments and address them accordingly. On top of that, I would like to suggest double-checking the following points and provide additional data, if possible.
The basal levels of TNF-alpha and IL-6 seem to be rather higher. Could you confirm it with other published articles?

Also, TNF-alpha and IL-6 were up-regulated by LPS treatment around 3-folds in comparison with the untreated. However, the windows between the untreated and the LPS-treated are rather narrow in the expression of MFGE8 and HMGN1. Have you tried in different concentrations of CM let alone 20 microM to see the dose-dependency?

In Figure 4C, the CM treatment resulted in the down-regulation in the expression of HMGN1 protein even lowered the expression in comparison to the untreated group. In this case, have you observed the decreased protein expression upon CM treatment in the untreated cells?

In Figures 5C and 5D, would it be possible to add si-NC, LPS+si-NC, LPS+si-MFGE8, and LPS+si-HMGN1 to clarify the effect of CM with LPS?

Lastly, because CM is an inhibitor of TMPRSS2 it would be great if authors postulate the link between TMPRSS2 and MFGE8 (or HMGN1) and add it to the discussion.

Reviewer 1 ·

Basic reporting

no comment

Experimental design

no comment

Validity of the findings

no comment

Additional comments

In summary
The authors reported that camostat mesilate inhibited IL-6 and TNF-α from DF-1 cells, as well as improved cell viability induced by LPS in vitro. They tried mass spectrometry to study the mechanism of camostat mesilate. They found MFGE8 and HMGN1 participated in IL-6 and TNF-α production and cell viability which regulated by camostat mesilate in DF-1 cells, and further confirmed them by using their siRNA.
The finding is based on in vitro data, it is interesting, but evidence is limited.
Specific points:
1. What is DF-1? You should specify this cell in manuscript and change the title.
2. It should be shown full name of MFGE8 and HMGN1 in abstract.
3. Figure 1. It should be mentioned statistic methods and numbers of samples you used in figure legend. Did you try different time points and concentration of CM?
4. Figure 2 and 3, I suggest you combine them, and should add your hypothesis about the results, especially related with your finding. In the legend, you should also specify the full name of DLE, DL, etc.
5. Figure 4, the concentration of LPS used here was so high, why did you choose this concentration? What is statistic method and numbers of samples you used?
6. Row 165-178, you should mention your MS data first, and then to clarify them. For MS data, people usually used western blot to confirm them, why did you choose RT-PCR?
7. Figure 5, you said si-HMGN1#2 was used for further experiment (Row 186), but in Figure 5C,D,E, you used si-HMGN1#1, which one is correct? Your current data were not enough to support your conclusion, controls (LPS alone, CM alone) should be added here. To support your hypothesis, it is better to use western blot to see the protein levels of MFGE8 and HMGN1 after treated with CM.
8. The English language should be improved to ensure that an international audience can clearly understand your text.

Reviewer 2 ·

Basic reporting

The authors have tested a known inhibitor of TMPRSS2 Camostat mesylate in chicken fibroblasts treated with LPS and illustrated that pro-inflammatory and cytokine secretion is decreased and the drug improves cell viability by regulating MFGE8 and HMGN1 in lipopolysaccharide stimulated DF-1 fibroblasts. The data is interesting but preliminary. The choice of fibroblast cell lines for this study is intriguing since they are not the main immune responders.

Experimental design

Specific comments
Provide specific references that chicken fibroblasts undergo TLR4 mediated signaling in response to LPS or bacteria. None of the references cited are of avian origin.
The TNF and IL6 ELISA kit specifications are not provided. These are necessary since the readers will want to know if the kits recognize chicken proteins.

Validity of the findings

In Fig 1C, there is not a drastic cell viability change in cells treated with LPS. Ideally a dose response to LPS and subsequent rescue by dosage of CM should be shown. The manuscript does mention the concentration of LPS was used in all assays. How did they determine this? Also, to assess pro inflammation and cell injury, more markers are needed.
What are the fold changes of the differentially expressed proteins selected for further analysis? Mention in the results. Also describe the functions of these proteins in the results itself.
In the qPCR, dose response to LPS/CM and also different time points for the RNA levels should be shown to convince the readers that these small changes are significant. This is a critical flaw. If antibodies are available, protein levels should be shown.
The characterization of SiRNAs can be removed to the supplementals.
Dose response is needed for Fig 5 to generate any conclusions that the authors put forward. The effect of LPS stimulation might be reversed after 72 hrs in some of these experiments.
If the target of CM is known, then how do the authors correlate the effects on these proteins? Discuss. It is easier to test these hypothesis on a human fibroblast cell line and another relevant cell line THP1 for eg; to show that these effects are true. Western blots can be easily performed in human cell lines with specific antibodies available.

Reviewer 3 ·

Basic reporting

The reporting is clear with some minor errors in the flow of the paper. The edits have been mentioned in the pdf. The literature use is accurate and the data is shown clearly to support the hypothesis

Experimental design

The experiments have been designed to support the hypothesis as well as to support the results obtained from the proteomic analysis obtained by the author using TMT.

Validity of the findings

The data have been validated in biological replicates and the supporting data adds to more robust underlying data which supports the findings reported in the manuscript

Additional comments

The authors have put together a relevant piece of work. The data is clearly shown and represented in the manuscript. There are some basic comments and concerns that have been addressed in the attached pdf.

Annotated reviews are not available for download in order to protect the identity of reviewers who chose to remain anonymous.

---

## Round 0.2 · accepted · Accept

Thank you for resubmitting the revised manuscript. The authors provide answers to the reviewers' comments and have revised the manuscript accordingly. Therefore, I consider it suitable for publication in its present form.

Reviewer 1 ·

Basic reporting

no comment

Experimental design

no comment

Validity of the findings

no comment

Additional comments

The authors have addressed all of my comments.

Reviewer 3 ·

Basic reporting

The authors have updated the manuscript based on the suggestions from the editor as well as the reviewers

Experimental design

The design has been updated

Validity of the findings

The manuscript is now focused and has significant clarity